# REALISTIC HUMAN MOTION GENERATION WITH CROSS-DIFFUSION MODELS

## ABSTRACT

We introduce the Cross Human Motion Diffusion Model (CrossDiff), a novel approach for generating high-quality human motion based on textual descriptions. Our method integrates 3D and 2D information using a shared transformer network within the training of the diffusion model, unifying motion noise into a single feature space. This enables cross-decoding of features into both 3D and 2D motion representations, regardless of their original dimension. The primary advantage of CrossDiff is its cross-diffusion mechanism, which allows the model to reverse either 2D or 3D noise into clean motion during training. This capability leverages the complementary information in both motion representations, capturing intricate human movement details often missed by models relying solely on 3D information. Consequently, CrossDiff effectively combines the strengths of both representations to generate more realistic motion sequences. In our experiments, our model demonstrates competitive state-of-the-art performance on text-to-motion benchmarks. Moreover, our method consistently provides enhanced motion generation quality, capturing complex full-body movement intricacies. Additionally, our approach accommodates using in the wild 2D motion data without 3D motion ground truth during training to generate 3D motion, highlighting its potential for broader applications and efficient use of available data resources.

## 1 INTRODUCTION

In recent years, the field of human motion synthesis (Li et al., 2017; Ghosh et al., 2017; Lee et al., 2019; Tseng et al., 2023; Yoon et al., 2020; Guo et al., 2022b; Li et al., 2021; Guo et al., 2020) has witnessed significant advancements, primarily driven by the growing demand for high-quality, realistic motion generation in applications such as gaming, virtual reality, and robotics. A crucial aspect in this research area is generating human motion based on textual descriptions, enabling contextually accurate and natural movements (Tevet et al., 2022b). However, current methods (Petrovich et al., 2022; Tevet et al., 2022b; Guo et al., 2022a; Chen et al., 2023b) predominantly rely on 3D motion information during training, leading to an inability to capture the full spectrum of intricacies associated with human motion. When using only 3D representation, the generation model may struggle to relate text semantics to some body part movements with very small movement variations compared to others, which can lead to overlooking important motion details. This is because the model might focus on more dominant or larger movements within the 3D space, leaving subtle nuances underrepresented. For example, when given a prompt such as "a person is dancing eloquently," as illustrated in Figure 1, the generated motion might lack vitality, display a limited range of movements, and contain minimal local motion details.

To effectively address the limitations and accurately capture the nuances of full-body movement, we introduce the Cross Human Motion Diffusion Model (CrossDiff). This innovative approach seamlessly integrates and leverages both 3D and 2D motion information to generate high-quality human motion sequences. The 2D data representation effectively illustrates the intricacies of human body movements from various viewing angle projections. Due to different view projections in 2D data, small body part movements can be magnified in certain projections, making them more noticeable and easier to capture. This helps the text-to-motion generation models to better associate text descriptions with a wider range of human body motion details, including subtle movements that might have been overlooked when relying solely on 3D representation.

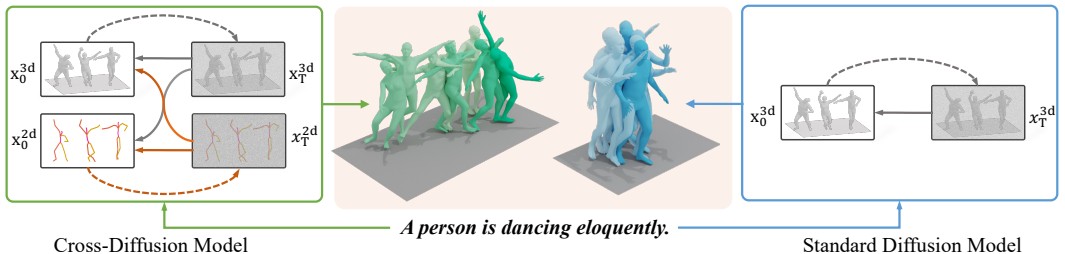

Figure 1: Our method utilizing the cross-diffusion mechanism (Left) exhibits more full-body details compared to existing methods (Right).

As a result, incorporating 2D information with 3D enables the diffusion model to establish more connections between motion and text prompts, ultimately enhancing the motion synthesis process. The CrossDiff learning framework consists of two main components: unified encoding and cross-decoding. These components work together to achieve more precise and realistic motion synthesis. Furthermore, it is essential to transfer the knowledge acquired in the 2D domain to 3D motion, which leads to an overall improvement in the model's performance.

Unified encoding fuses motion noise from both 3D and 2D sources into a single feature space, facilitating cross-decoding of features into either 3D or 2D motion representations, regardless of their original dimension. The distinctive innovation of our approach stems from the cross-diffusion mechanism, which enables the model to transform 2D or 3D noise into clean motion during the training process. This capability allows the model to harness the complementary information present in both motion representations, effectively capturing intricate details of human movement that are often missed by models relying exclusively on 3D data.

In experiments, we demonstrate our model achieves competitive state-of-the-art performance on several text-to-motion benchmarks, outperforming existing diffusion-based approaches that rely solely on 3D motion information during training. Furthermore, our method consistently delivers enhanced motion generation quality, capturing complex full-body movement intricacies essential for realistic motion synthesis. A notable advantage of our approach is its ability to utilize 2D motion data without necessitating 3D motion ground truth during training, enabling the generation of 3D motion. This feature underscores the potential of the CrossDiff model for a wide range of applications and efficient use of available data resources.

## 2 RELATED WORK

### 2.1 HUMAN MOTION GENERATION

Human motion generation is the process of synthesizing human motion either unconditionally or conditioned by signals such as text, audio, or action labels. Early works (Li et al., 2017; Ghosh et al., 2017; Pavllo et al., 2018; Li et al., 2020) treated this as a deterministic mapping problem, generating a single motion from a specific signal using neural networks. However, human motion is inherently stochastic, even under certain conditions, leading to the adoption of deep generative models in more recent research.

For instance, Dancing2music (Lee et al., 2019) employed GANs to generate motion under corresponding conditions. ACTOR (Petrovich et al., 2021) introduced a framework based on transformers (Vaswani et al., 2017) and VAEs, which, although designed for action-to-motion tasks, can be easily adapted for text-to-motion tasks as demonstrated in TEMOS (Petrovich et al., 2022). Since text and audio are time-series data, natural language processing approaches are commonly used. Works by Ghosh et al. (2021), Ahuja & Morency (2019), and Guo et al. (2022a) utilized GRU-based language models to process motion data along the time axis.

Tevet et al. (2022a) developed MotionCLIP, which uses the shared text-image space learned by CLIP (Radford et al., 2021) to align the feature space of human motion with that of CLIP. MotionGPT (Sohl-Dickstein et al., 2015; Song & Ermon, 2020) directly treats motion as language and

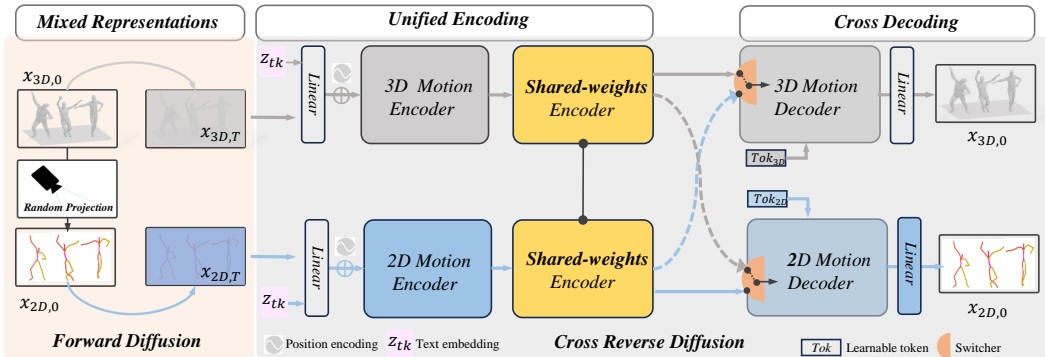

Figure 2: Overview of our CrossDiff framework for generating human motion from textual descriptions. The framework incorporates both 3D and 2D motion data, using unified encoding and cross-decoding components to process mixed representations obtained from random projection.

addresses the motion generation task as a translation problem. However, conditions like language and human motion differ significantly in terms of distribution and expression, making accurate alignment challenging.

To overcome this issue, T2M-GPT (Zhang et al., 2023a) and TM2T (Guo et al., 2022b) encode motion using VQ-VAE (Van Den Oord et al., 2017) and generate motion embeddings with generative pretrained transformers. Zhang et al. (2022) introduced MotionDiffuse, the first application of diffusion models in text-to-motion tasks. Tevet et al. (2022b) presented MDM, which employs a simple diffusion framework to diffuse raw motion data, while Chen et al. (2023b) proposed MLD, which encodes motion using a VAE model and diffuses it in the latent space. ReMoDiffuse (Zhang et al., 2023b) retrieves the motion related to the text to assist in motion generation. Meanwhile, Fg-T2M (Wang et al., 2023) utilizes a fine-grained method to extract neighborhood and overall semantic linguistic features. Although these methods attain success, they depend exclusively on 3D motion data during training, which results in a failure to capture sufficient complexities associated with human motion. In contrast, our approach utilizes a cross-diffusion mechanism to leverage the complementary information found in both 2D and 3D motion representations.

## 2.2 DIFFUSION MODELS

Diffusion generative models (Sohl-Dickstein et al., 2015; Song & Ermon, 2020; Ho et al., 2020), based on stochastic diffusion processes in Thermodynamics, involve a forward process where samples from the data distribution are progressively noised towards a Gaussian distribution and a reverse process where the model learns to denoise Gaussian samples. These models have achieved success in various domains, including image synthesis (Saharia et al., 2022; Ramesh et al., 2022; Rombach et al., 2022; Sinha et al., 2021; Vahdat et al., 2021), video generation (Ho et al., 2020; Yang et al., 2022; Luo et al., 2023), adversarial attacks (Zhuang et al., 2023; Nie et al., 2022), motion prediction (Wei et al., 2023; Chen et al., 2023a), music-to-dance synthesis (Li et al., 2023; Tseng et al., 2023), and text-to-motion generation (Zhang et al., 2022; Tevet et al., 2022b; Chen et al., 2023b; Ren et al., 2023; Yuan et al., 2022).

## 3 METHOD

### 3.1 OVERVIEW

Given a textual description $c$, our objective is to generate multiple human motion sequences $x^{1:N} = \{x^i\}_{i=1}^{N}$, each with a length of $N$. As illustrated in Figure 2, our method is carefully designed to efficiently incorporate both 3D and 2D motion data within the learning process of the diffusion model.

During the training phase, we first obtain mixed representations of the data from the provided 3D input using a random projection technique. Afterward, the 2D and 3D data representations are independently diffused and processed through our learning framework, CrossDiff, which primarily consists of unified encoding and cross-decoding components.

The unified encoding module maps both the 2D and 3D data into a shared feature space. These features are then passed through the cross-decoding component, resulting in the generation of two motion representations. These representations are subsequently employed for loss calculation and model learning. In the inference phase, our approach supports not only standard sampling but also mixture sampling.

**Preliminary.** Denoising diffusion probabilistic models (DDPM) (Ho et al., 2020) can iteratively eliminate noise from a gaussian distribution to approximate a true data distribution. This technique has had a significant impact on the field of generative research, including text-to-motion applications. In this study, we have adapted DDPM and trained a transformer-based model to gradually reduce noise and generate motion sequences.

Diffusion is modeled as a Markov noising process $\{x_t^{1:N}\}_{t=0}^T$ of $T$ steps. For simplicity, we use $x_t$ to denote $x_t^{1:N}$ in the following discussion. Starting with a motion sequence $x_0$ in original data distribution, the noising process can be described as

$$q(x_t|x_{t-1}) = \mathcal{N}(x_t; \sqrt{\alpha_t}x_{t-1}, (1-\alpha_t)\mathbf{I}) \tag{1}$$

where $\alpha_t \in (0,1)$ is constant hyper-parameters. When $\alpha_T$ is small enough, we can approximate $x_T \in \mathcal{N}(0,\mathbf{I})$. The reverse process is to progressively denoise $x_T$ from a gaussian distribution to obtain the clean motion $x_0$. Following Ramesh et al. (2022); Tevet et al. (2022b), we predict the clean motion $x_0$ itself on textual condition $c$ as $\hat{x}_0 = G(x_t, t, c)$. We apply the simple objective (Ho et al., 2020)

$$\mathcal{L}_{simple} = \mathbb{E}_{t\sim[1,T]}||x_0 - G(x_t, t, c)||_2^2. \tag{2}$$

### 3.2 MIXED REPRESENTATIONS

As the naive diffusion model is trained only on one data distribution (3D poses), we have trained it on a mixture representation of 3D and 2D poses. To obtain 2D data that is closer to the real distribution, we randomly projected the 3D poses into 2D planes in four directions (front, left, right, and back). The 3D poses $x_{3D}^i \in \mathbb{R}^{d_{3D}}$ and 2D poses $x_{2D}^i \in \mathbb{R}^{d_{2D}}$ are represented respectively by $d_{3D}$-dimensional and $d_{2D}$-dimensional redundant features, respectively, as suggested by Guo et al. (2022a). The pose $x^i$ is defined by a tuple of $(r, j^p, j^v, j^r, c^f)$, where $(r_{3D}, j_{3D}^p, j_{3D}^v, j_{3D}^r, c_{3D}^f)$ is identical to Guo et al. (2022a). In addition, $r_{2D} \in \mathbb{R}^2$ represents 2D root velocity. $j_{2D}^p \in \mathbb{R}^{2j}, j_{2D}^v \in \mathbb{R}^{2j}$ and $j_{2D}^r \in \mathbb{R}^{2j}$ represent the local joints positions, velocities and rotations, respectively, with $j$ denoting the number of joints besides the root. $c_{2D}^f \in \mathbb{R}^4$ is a set of binary features obtained by thresholding the heel and toe joint velocities. Notably, the rotation representation is made up of the sine and cosine values of the angle.

### 3.3 CROSS MOTION DIFFUSION MODEL

**Framework.** Our pipeline is illustrated in Figure 2. CLIP (Radford et al., 2021) is a widely recognized text encoder, and we use it to encode the text prompt $c$. The encoded text feature and time-step $t$ are projected into transformer dimension and summed together as the condition token $z_{tk}$. The 2D and 3D motion sequences are projected into the same dimension, concatenated with condition token $z_{tk}$ in time axis and summed with a standard positional embedding. We aim to unify the two domain features in one space but it is too difficult for one linear layer. A straightforward idea is to encode via two separate encoders:

$$z_{3D}^0 = \mathcal{E}_{3D}(x_{3D,t}, t, c), z_{2D}^0 = \mathcal{E}_{2D}(x_{2D,t}, t, c), \tag{3}$$

where $\mathcal{E}_{3D/2D}(\cdot)$ are 3D/2D $L_1$-layer transformer encoders (Vaswani et al., 2017). However, We find it more efficient to add another shared-weight encoder to extract shared feature:

$$\{z_{3D/2D}^i\}_{i=1}^{L_2} = \mathcal{E}_{share}(z_{3D/2D}^0), \tag{4}$$

where $\mathcal{E}_{share}(\cdot)$ is a shared-weight $L_2$-layer transformer encoder, and $\{z^i_{3D/2D}\}^{L_2}_{i=1}$ are the outputs of each shared-weight layer. The whole process is defined as unified encoding.

To output motion in two modality, we use independent $L_2$-layer transformer decoders (Vaswani et al., 2017) for 2D and 3D data. Starting from 2D/3D learnable token embeddings $Tok_{2D/3D}$, each decoder layer takes the output of the previous layer as queries and the output of same-level shared layer $z^i$ as keys and values instead of the last layer. The starting point is to make decoder layers follow the extracting patterns of shared-weight layers rather than gaining deeper embeddings. Finally, a linear layer is added to map the features to the dimensions of the motions. This cross-decoding can be integrated as:

$$\hat{x}_{3D,0} = \mathcal{D}_{3D}(\{z^i_{3D/2D}\}^{L_2}_{i=1}), \hat{x}_{2D,0} = \mathcal{D}_{2D}(\{z^i_{3D/2D}\}^{L_2}_{i=1}), \tag{5}$$

where $\mathcal{D}_{3D/2D}$ are 3D/2D decoders including learnable tokens; and $\hat{x}_{3D/2D,0}$ are predicted clean 3D/2D motion sequences. In summary, with a pair of 3D motion and 2D projected motion, CrossDiff outputs four results via

$$\hat{x}_{iD \to jD,0} = G_{iD \to jD}(x_{iD,t}, t, c) = \mathcal{D}_{jD}(\mathcal{E}_{share}(\mathcal{E}_{iD}(x_{iD,t}, t, c))), \tag{6}$$

where $\hat{x}_{iD \to jD,0}$ are predicted $j$-dimension clean motion $\hat{x}_{jD,0}$ from $i$-dimension motion noise $x_{iD,t}$ with $i, j \in \{2, 3\}$.

**Training.** As mentioned in Section 3.1, we apply a simple objective (Eq. 2) for all outputs:

$$\mathcal{L}_{iD \to jD} = \mathbb{E}_{t \sim [1,T]}||x_{jD,0} - G_{iD \to jD}(x_{iD,t}, t, c)||^2_2. \tag{7}$$

We train our model in two stage. In stage I, CrossDiff is forced to learn the reverse process, motion characteristic of texts in both domains and the motion connection of two distribution via the loss

$$\mathcal{L}_{stageI} = \mathcal{L}_{3D \to 3D} + w_{23}\mathcal{L}_{2D \to 3D} + w_{32}\mathcal{L}_{3D \to 2D} + w_{22}\mathcal{L}_{2D \to 2D}, \tag{8}$$

where $w_{23}, w_{32}, w_{22}$ are relative weights. In stage II, there is only a 3D generation loss:

$$\mathcal{L}_{stageII} = \mathcal{L}_{3D \to 3D}. \tag{9}$$

This helps the model focus on the 3D denoising process and eliminate the uncertainty of the 2D and 3D mapping relationship while retaining the knowledge of diverse motion features.

## 3.4 Mixture Sampling

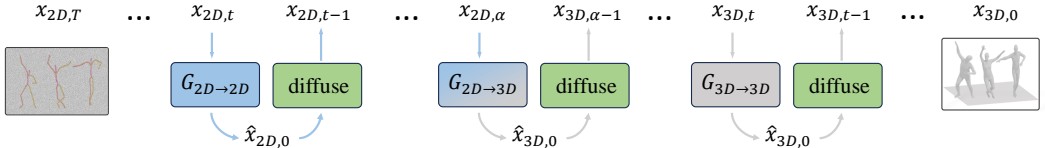

Figure 3: **Overview of Mixture Sampling.** The original noise is sampled from a 2D gaussian distribution. From time-step $T$ to $\alpha$, CrossDiff predicts the clean 2D motion $\hat{x}_{2D,0}$ and diffuses it back to $x_{2D,t-1}$. In the remaining $\alpha$ steps, CrossDiff denoises in the 3D domain and finally obtains the clean 3D motion.

After training, one can sample a motion sequence conditioned on a text prompt in an iterative manner. The standard method (Tevet et al., 2022b) gradually anneals the 3D noise from a gaussian distribution, which we still use. We predict the clean sample $\hat{x}_{3D,0}$ and noise it back to $x_{3D,t-1}$ for $T$ steps until $x_{3D,0}$.

Furthermore, utilizing the CrossDiff architecture, we propose a novel two-domain sampling approach. As shown in Figure 3, We first sample 2D gaussian noise which is then denoised with $G_{2D \to 2D}(x_{2D,t}, t, c)$ until time-step $\alpha$. Next, we project the denoised 2D noise onto the 3D domain using $G_{2D \to 3D}(x_{2D,t}, t, c)$ and continue the denoising process with $G_{3D \to 3D}(x_{3D,t}, t, c)$ for the remaining $\alpha$ steps. Our experiments in Appendix B demonstrate the difference between mixture sampling and the vanilla method.

## 3.5 LEARNING 3D MOTION GENERATION FROM 2D DATA

Given the complexity and cost associated with collecting high-quality 3D motion data, generating 3D motion from 2D motion data is an attractive alternative. Moreover, generating 3D motion from textual descriptions in an out-of-domain scenario is approximately a zero-shot task. To achieve this, we first estimated 2D motion from text-related videos using an off-the-shelf model. We then utilized the pretrained model in stage I $G_{2D \rightarrow 3D}(x_{2D,t}, t, c)$ to generate corresponding 3D clean motion, with $t$ set to 0 and $c$ set to null condition $\emptyset$. A motion filter is applied to smooth the generated motion. We assume that 2D pose estimation is relatively precise, allowing the processed 2D/3D motion to serve as pseudo-labels for training. The model is fine-tuned with the same objective as stage I, but with different weight hyper-parameters. After training, our model can generate diverse motion according to out-of-domain textual descriptions using mixture sampling (Sec. 3.4).

During training with 2D motion estimated from videos, we encounter significant errors in root estimation due to the uncertainty of camera movement. To address this issue, we decouple the root information $r_{3D/2D}$ and generate it based on other pose features. Please refer to Appendix C for more details.

# 4 EXPERIMENTS

## 4.1 DATASETS AND EVALUATION METRICS

**Datasets.** We evaluate on HumanML3D (Guo et al., 2022a) and KIT-ML datasets (Plappert et al., 2016) and convert 3D motion to 2D using orthogonal projection and process the redundant 2D motion representation as outlined in Section 3.2. For 2D data training experiments, we use the UFC101 dataset (Soomro et al., 2012), extracting 2D joint positions with the ViTPose model (Xu et al., 2022). After filtering unclear data, we select 24 action labels with 10-50 motion sequences each and annotate five textual descriptions per label.

**Evaluation Metrics.**

We compare our results with previous works using the same metrics as (Guo et al., 2022a; Tevet et al., 2022b). These metrics involve evaluating quality with Frechet Inception Distance (**FID**), precision with **R-Precision** and Multi-modal Distance (**MM Dist**), and diversity with Diversity (**DIV**) and Multimodality (**MModality**).

Besides using the evaluation model from Guo et al. (2022a), we introduce a new metric measuring the FID (Fréchet Inception Distance) of upper and lower body movements, denoted as **FID-U** and **FID-L**. This enables a fine-grained analysis of human motion and better comprehension of upper and lower body dynamics. We split joints into two groups using the root joint as a boundary and train separate evaluators, following a similar approach to Guo et al. (2022a). This effectively evaluates generated motion quality for both body segments, offering a deeper understanding of human motion complexities and advancing research on new motion generation models.

Table 1: Quantitative results on the HumanML3D and KIT-ML test set. The overall results on KIT-ML are shown on the **right**, while the results of both widely-used and newly-proposed metrics on HumanML3D are shown on the **left**. The red and blue colors indicate the best and second-best results, respectively.

| Methods | HumanML3D | | | | | | | KIT-ML | | | | |
| --- | --- | --- | --- | --- | --- | --- | --- | --- | --- | --- | --- | --- |
| | R Precision (top 3)↑ | FID↓ | MM Dist↓ | DIV→ | MModality↑ | FID-U↓ | FID-L↓ | R Precision (top 3)↑ | FID↓ | MM Dist↓ | DIV→ | MModality↑ |
| Real | 0.797 | 0.002 | 2.974 | 9.503 | - | - | - | 0.779 | 0.031 | 2.788 | 11.08 | - |
| Language2Pose | 0.486 | 11.02 | 5.296 | 7.676 | - | - | - | 0.483 | 6.545 | 5.147 | 9.073 | - |
| T2G | 0.345 | 7.664 | 6.030 | 6.409 | - | - | - | 0.338 | 12.12 | 6.964 | 9.334 | - |
| Hier | 0.552 | 6.532 | 5.012 | 8.332 | - | - | - | 0.531 | 5.203 | 4.986 | 9.563 | - |
| T2M | 0.740 | 1.067 | 3.340 | 9.188 | 2.090 | - | - | 0.693 | 2.770 | 3.401 | 10.91 | 1.482 |
| MotionDiffuse | 0.782 | 0.630 | 3.113 | 9.410 | 1.553 | - | - | 0.739 | 1.954 | 2.958 | 11.10 | 0.730 |
| ReMoDiffuse | 0.795 | 0.103 | 2.974 | 9.018 | 1.795 | - | - | 0.765 | 0.155 | 2.814 | 10.80 | 1.239 |
| Fg-T2M | 0.783 | 0.243 | 3.109 | 9.278 | 1.614 | - | - | 0.745 | 0.571 | 3.114 | 10.93 | 1.019 |
| MDM | 0.611 | 0.544 | 5.566 | 9.559 | 2.799 | 0.825 | 0.840 | 0.396 | 0.497 | 9.191 | 10.847 | 1.907 |
| T2M-GPT | 0.775 | 0.141 | 3.121 | 9.722 | 1.831 | 0.145 | 0.607 | 0.745 | 0.514 | 3.007 | 10.921 | 1.570 |
| MLD | 0.772 | 0.473 | 3.196 | 9.724 | 2.413 | 0.541 | 0.553 | 0.734 | 0.404 | 3.204 | 10.80 | 2.192 |
| Ours | 0.730 | 0.162 | 3.358 | 9.577 | 2.620 | 0.118 | 0.281 | 0.704 | 0.474 | 3.308 | 10.77 | 1.742 |

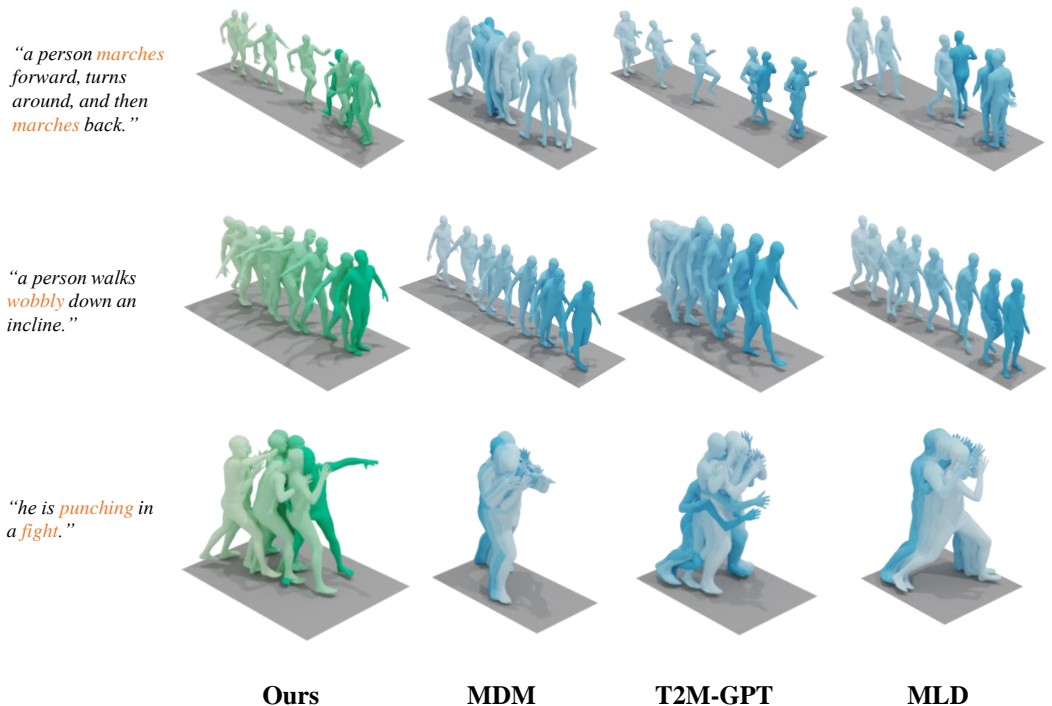

"a person marches forward, turns around, and then marches back."

"a person walks wobbly down an incline."

"he is punching in a fight."

**Ours**          **MDM**          **T2M-GPT**          **MLD**

Figure 4: Qualitative results on HumanML3D dataset. We compare our method with MDM (Tevet et al., 2022b), T2M-GPT (Zhang et al., 2023a) and MLD (Chen et al., 2023b). We find that our generated actions better convey the intended semantics.

## 4.2 COMPARISONS ON TEXT-TO-MOTION

**Comparative Analysis of Standard Metrics.** In our evaluation, we test our models 20 times and compare their performance with existing state-of-the-art methods. These methods include Language2Pose (Ahuja & Morency, 2019), T2G (Bhattacharya et al., 2021), Hier (Ghosh et al., 2021), T2M (Guo et al., 2022a), MotionDiffuse (Zhang et al., 2022), ReMoDiffuse (Zhang et al., 2023b), Fg-T2M (Wang et al., 2023), MDM (Tevet et al., 2022b), T2M-GPT (Zhang et al., 2023a), and MLD (Chen et al., 2023b). As illustrated in Table 1, our model exhibits competitive performance when compared to these leading methods. However, it is important to note that the KIT-ML dataset primarily consists of "walk" movements and lacks intricate details. Consequently, this dataset does not present the same challenges that our method is specifically designed to address.

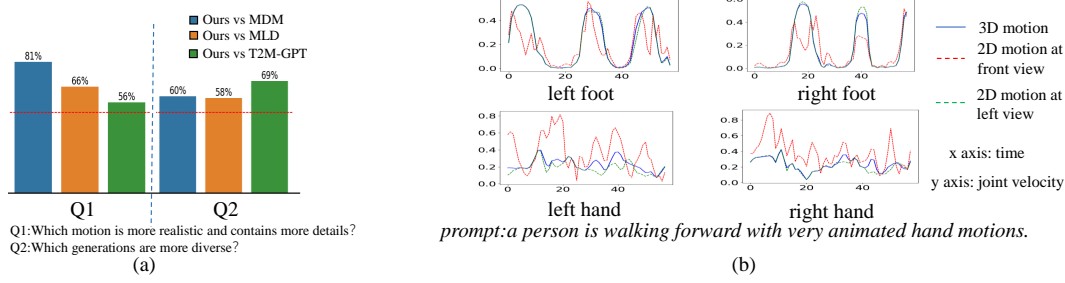

Figure 5: (a)The result of the user study. (b) Difference between 3D and 2D motion data distribution. The time axis is represented on the x-axis, while the normalized joint velocity is represented on the y-axis. The 3D motion is represented by a blue full line, while the 2D motion is represented by red and green dashed lines, indicating the front and left view, respectively.

**Comparative Analysis of Fine-grained Metrics.** We compare the fine-grained metrics for our upper and lower body with those from three recent studies (Tevet et al., 2022b; Zhang et al., 2023a; Chen et al., 2023b). As demonstrated in Table 1, our generated motion is more robust and detailed. Our low FID scores for both the upper and lower body indicate that our synthesized motion effectively captures full-body movement rather than focusing solely on specific semantic parts. In contrast, T2M-GPT achieves a low FID score for the upper body but a high score for the lower body. This suggests that their generation process exhibits unbalanced attention towards different body parts, primarily translating textual descriptions into upper body characteristics rather than capturing the entire body's motion.

Figure 4 displays qualitative comparisons with existing methods. Our method can "march" with arm swings, "wobble" using hands for balancing and alternate between defense and attack in a "fight". We conducted a user study on motion performance, in which participants were asked two questions to assess the vitality and diversity of the motions. The results, presented in Figure 5(a), confirm our analysis. Detail information is in Appendix D. In summary, our method demonstrates a superior ability to interpret semantic information and generate more accurate and expressive motions.

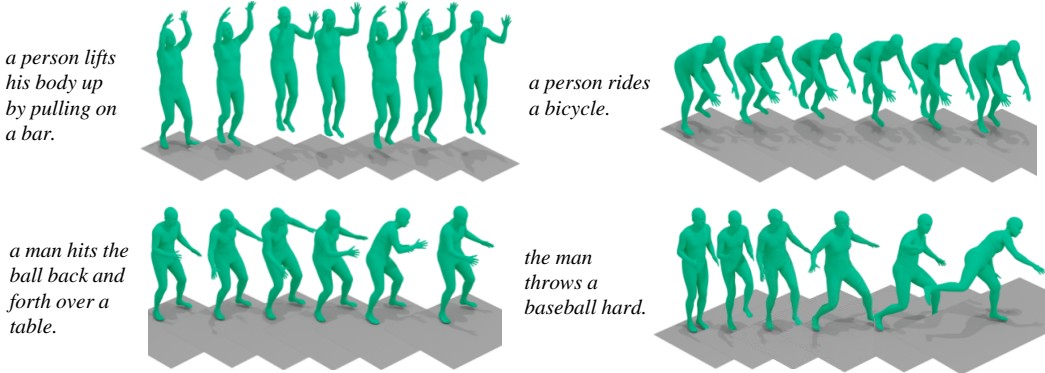

Figure 6: Generating 3D movements without training on paired 3D motion and textual descriptions.

## 4.3 LEARNING FROM 2D DATA

After being trained on a 3D dataset, our model can learn 3D motion generation from 2D data. By fine-tuning the model with the UCF101 dataset (Soomro et al., 2012), we effectively address a zero-shot problem arising from the absence of ground-truth 3D motion. Our sampling strategy reaches optimal performance when $\alpha = 1$. As depicted in Figure 6, the generated motions for various activities, such as pulling up, biking, table tennis, and baseball, are showcased alongside their textual prompts. Despite some activities being beyond the scope of the HumanML3D domain, our fine-tuned model successfully synthesizes specific motions by leveraging the weak 2D data. This demonstrates its remarkable adaptability and potential for efficient use of available motion data.

## 4.4 ABLATION STUDIES

Our model features separate pipelines for 2D and 3D inputs, allowing us to train solely on 3D motion sequences, which is an improvement over MDM (Tevet et al., 2022b). We investigate the impact of introducing 2D data on the performance of 3D generation and demonstrate the effectiveness of using a shared-weights encoder.

### 4.4.1 WHY 2D MOTION HELP?

To explain the benefits of 2D motions, we compared the distribution differences between 3D and 2D motion data. Hand and feet movements, which are primary indicators of motion, were visualized in both 3D and 2D levels, and their velocities were normalized along the joints dimension. The results in Figure 5(b) show that 2D motion captures different details from 3D motion, indicating that the CrossDiff model can lead 3D motion to learn from the knowledge that 2D motion acquired from

Table 2: Evaluation of our models with different settings on the HumanML3D dataset. **Bold** indicates best result. The symbol % indicates the percentage of data being used. From top to bottom, we present MDM as baselines, the impact of training with 2D representations, with(w/) or without(w/o) shared-weights encoder.

| Methods | R Precision (top 3)↑ | FID↓ | MM Dist↓ | DIV→ |
|---------|:---:|:---:|:---:|:---:|
| MDM | 0.611 | 0.544 | 5.566 | 9.559 |
| Ours | **0.730** | **0.162** | **3.358** | **9.577** |
| 50% 3D | 0.666 | 0.586 | 3.894 | 9.513 |
| 100% 3D | 0.685 | 0.224 | 3.690 | 9.445 |
| 50% 3D + 100% 2D | 0.672 | 0.422 | 3.708 | 9.345 |
| 100% 3D + 100% 2D | 0.730 | 0.162 | 3.358 | 9.577 |
| w/o shared-weights encoder | 0.714 | 0.187 | 3.496 | 9.488 |
| w/ shared-weights encoder | 0.730 | 0.162 | 3.358 | 9.577 |

text prompts. Specifically, for the given sample, 2D motion might learn "animated hand motions" while 3D motion focuses only on "walking". 2D motion is an explicit feature that we artificially extract to aid the model's learning, and this approach can help improve performance when dealing with arbitrary data.

### 4.4.2 INFLUENCE OF 2D REPRESENTATION

Table 2 presents the results from four different experiment settings. The control groups of "100% 3D" and "100% 3D + 100% 2D" demonstrate that when training with paired 3D motion and text, projecting the 3D motion to 2D and building a connection between the 2D motion and text can help boost performance. The visualizations in Figure 1 further highlight the enhanced quality of our generated outputs. The control groups of "50% 3D" and "50% 3D + 100% 2D" prove that additional 2D data can also help improve performance. The additional 2D data indicates other 2D motion without ground truth 3D motion. The experiment in Section 4.3 shows learning 2D motion in the wild can also help with out-of-domain 3D motion learning. As we can see, the combined learning of 2D motion has great potential.

### 4.4.3 SHARED-WEIGHTS ENCODER

Without the shared-weights encoder, the model is a simple encoder-decoder framework with two modalities. However, we believe that this is not sufficient to fully fuse the 3D and 2D motion features. Inspired by Xu et al. (2023), we found that when learning with data from two modalities, extracting separate and fused feature layer-by-layer is more efficient. The shared-weights encoder serves as a fusing module, while the 3D/2D motion decoder acts as a separate decoder module. The goal is to ensure that the decoder layers follow the same extraction patterns as the shared-weight layers, rather than simply gaining deeper embeddings. The results presented in Table 2 demonstrate the efficiency of using a shared-weights encoder.

## 5 CONCLUSION

In conclusion, the Cross Human Motion Diffusion Model (CrossDiff) presents a promising advancement in the field of human motion synthesis by effectively integrating and leveraging both 3D and 2D motion information for high-quality motion generation. The unified encoding and cross-decoding components of the CrossDiff learning framework enable the capture of intricate details of human movement that are often overlooked by models relying solely on 3D data. Our experiments validate the superior performance of the proposed model on various text-to-motion benchmarks. Future work could explore methods to enhance the model's generalization capabilities, such as incorporating unsupervised or semi-supervised learning techniques.

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

# A    2D REPRESENTATION FROM DIFFERENT VIEWS

Table 3: The differences among 2D Representation from different views.

| Methods | R Precision (top 3)↑ | FID↓ | MM Dist↓ | DIV→ |
|---|---|---|---|---|
| 1 view(front) | 0.722 | 0.186 | 3.467 | 9.798 |
| 1 view(left) | 0.715 | 0.181 | 3.412 | 9.834 |
| 4 views | 0.730 | 0.162 | 3.358 | 9.577 |
| 5 views | 0.695 | 0.202 | 3.613 | 9.502 |

To evaluate the impact of different views, we conducted tests on four settings: a) only the front view, b) only the left view, c) four views including front, left, back, and right, and d) five views with an additional top view. When multiple views were used, the 3D motion was paired with a random view projection to formulate the loss. As shown in Table 3, it is reasonable to assume that four views contain more 2D information and outperform one view. Furthermore, the front view and left view do not have much distinction in terms of performance. However, the performance actually decreased with the additional top view. This could be due to the fact that the 2D information becomes more difficult to classify without the condition of the camera view. We did not pass along the camera information because it is difficult to estimate the camera view of in-the-wild videos, and injecting camera information could disrupt the structure of the text-to-2D model, making it difficult for the text-to-3D model to follow. In conclusion, the number of views serves as a practical hyperparameter that can be adjusted through enumeration experiments.

# B    COMPARISON OF SAMPLING METHOD

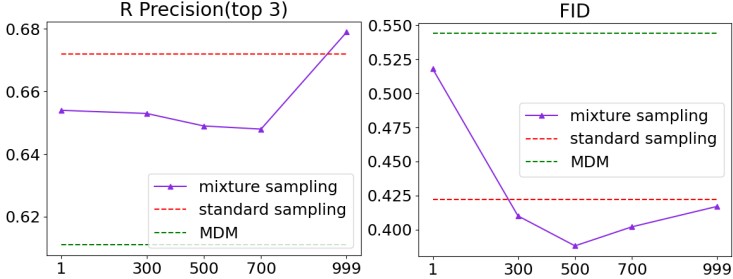

Figure 7: Results of different sampling strategies are presented in terms of R Precision (top 3) and FID. The x-axis represents the number of steps $\alpha$ for 3D denoising. The red dashed line represents denoising a 3D noise only in the 3D domain, which is the standard sampling method. The green dashed line represents the results of MDM.

In Section 3.4, we discuss how our model can denoise and reconstruct 3D movements from both 3D and 2D noise. To investigate if incorporating more 2D data can improve generation performance, we evaluate different sampling methods. For this experiment, we train our model using 50% 3D motion and all available 2D motion, and diffuse 1k steps in all methods. The parameter $\alpha$ indicates the time-step at which the motion representation switches from 2D to 3D. The red dashed line represents the standard sampling method.

The results, as shown in Figure 7, reveal that $\alpha = 500$ achieves the lowest FID score, indicating that our model can effectively use abundant 2D motion to enhance 3D generation performance. However, the standard sampling method scores high on R-Precision, as the generation is more precise. Additional results are provided in the supplementary material.

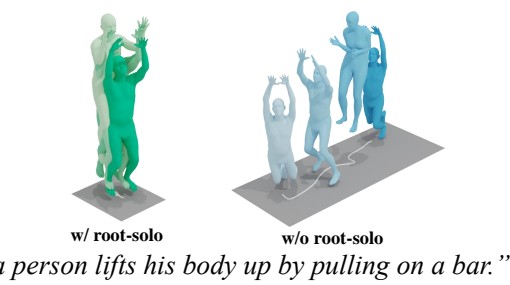

w/ root-solo        w/o root-solo

*"a person lifts his body up by pulling on a bar."*

Figure 8: Visualization of generated motions with and without the root-decoupled diffusion model.

## C   EFFECT OF ROOT-DECOUPLED DIFFUSION

To address the uncertainty of camera movement for in-the-wild videos, we decouple the root information $r_{3D/2D}$ and generate it based on other pose features. To achieve this, we employ a $L_3$-layer transformer encoder to encode the root information, which is then decoded with a $L_4$-layer transformer decoder conditioned on the last $L_4$ layer 3D/2D decoder outputs. $L_3$ and $L_4$ are set to 2 in experiments.

The comparison of generating with and without this technique is illustrated in Figure 8. Without root-decoupled diffusion, the generated motion exhibits random foot sliding while performing pulling-ups, as it learns from 2D motion data. However, since 2D motion sequences captured from videos may not accurately capture the root position, generating global movement based on body pose can lead to more precise results.

## D   DETAILS OF USER STUDY

We ask two questions in user studies to assess the vitality and diversity of the motions. The first is "Which motion is more realistic and contains more details?" and the participant is given a generated motion of our method and the compared method to choose. The second is "Which generations are more diverse?" and the participant is given three generated motions of our method and the compared method to choose. We eventually received 135 feedbacks. Considering the response time under 1 minute is invalid, we finally collate 113. Figure 5(a) shows that most of the time, CrossDiff was preferred over the compared models.

