# OpenReview forum: "Realistic Human Motion Generation with Cross-Diffusion Models"
_ICLR.cc/2024/Conference — Submitted to ICLR 2024_

### Official Review · Reviewer_kGCp · 2023-10-22

**Soundness:** 2 fair
**Presentation:** 3 good
**Contribution:** 2 fair
**Rating:** 5
**Confidence:** 3

**Summary:**

This work proposes to leverage both 3D and 2D information (by projecting 3D data into 2D) for 3D human motion generation, in two stages: unified encoding and cross decoding.

**Strengths:**

- I would like to commend that the proposed framework could leverage 2D data in training 3D motion generators (Table 2), I think this direction has immense potential. I would even encourage the authors to explore the relationship between the proposed framework and lifting-based 3D pose estimation.
- The paper is clearly presented, with helpful illustrations.

**Weaknesses:**

- The major weakness of this work is the lack of explanation for why 2D information could help in 3D human motion generation, given 2D motions are merely a projection of 3D motions onto four orthogonal views. Such projection only reduces the 3D information, without introducing new information. Unfortunately, there is no convincing theoretical motivation behind such an operation. Specifically, "... complementary information in both motion representations, capturing intricate human movement details often missed by models relying solely on 3D information", would you explain precisely how the projection helps provide "intricate" details, which are "complementary" to 3D information, given that they come from 3D in the first place?

- More analysis of 2D information would be helpful. For example, which view from the four (front, left, right, and back) is the most useful? Would a top view be helpful too? I feel the current version creates more questions than it answers.

- Considering the losses, is the framework aware of the input/target view (front, left, right, and back)? Specifically, for each 3D motion, how is the four 2D projection paired in the training? Some more details would be helpful.

- I wonder if the $x_{2D}$ -> $\hat{x}_{3D}$ motion generation is linked to 3D pose estimation via lifting (such as [A] and many follow-up works)?

- Experiment results are not very competitive in Table 1. However, I do not consider this a significant weakness as I recognize the proposed method's potential.

- A video in the supp would be helpful, as "high-quality" motion generation has been mentioned in the manuscript.

- What is the concrete conclusion we could draw from Figure 6a)? Mixing sampling performs better in R Precision with a large $\alpha$, but consistently outperformed by the standard sampling in terms of FID? It is very common to have conflicting trends with different metrics, but some more elaboration will be helpful.

- Minor: Figure 6 will benefit from some reformatting. Currently, the figure is too small, while large margins waste a lot of space.

- Minor: it would be hard to consider the root-decoupled diffusion as a significant novelty.

[A] Martinez et al., A simple yet effective baseline for 3D human pose estimation, ICCV'17

**Questions:**

Please refer to the weaknesses section.

---

> ### Author Response · Authors · 2023-11-17
> **Authors Response to Reviewer kGCp[1/2]**
>
> We appreciate your enlightening comments, which have facilitated our deeper thinking. Below are our explanations and thoughts on your concerns:
>
> **Q1: How does the projection help provide "intricate" details?**
>
> As we have mentioned in **Response to All Reviewers**, When using only 3D representation, the generation model may struggle to relate text semantics to some body part movements with very small movement variations compared to others, which can lead to overlooking important motion details. This is because the model might focus on more dominant or larger movements within the 3D space, leaving subtle nuances underrepresented. Due to different view projections in 2D data, small body part movements can be magnified in certain projections, making them more noticeable and easier to capture. This helps the text-to-motion generation models to better associate text descriptions with a wider range of human body motion details, including subtle movements that might have been overlooked when relying solely on 3D representation.
>
> To provide a more specific explanation, we have included a complete analysis of 2D and 3D motion in Section 4.4.1, which should help in understanding how the projection helps provide "intricate" details. It is the difference in the distribution of 2D and 3D motion data that inspired us to learn the relation between 2D motion and text.
>
> **Q2: More analysis of the 2D information.**
>
> To evaluate the impact of different views, we conducted tests on four settings: (a) only the front view, (b) only the left view, (c) four views including front, left, back, and right, and (d) five views with an additional top view. When multiple views were used, the 3D motion was paired with a random view projection to formulate the loss. The results is shown below:
>
> | Methods | R Precision(top 3) $\uparrow$ | FID $\downarrow$ | MM Dist $\downarrow$ | DIV $\rightarrow$ |
> | ----- | ------ | ----- | ----- | ---- |
> |1 view(front) | 0.722 | 0.186 | 3.467 | 9.798 |
> |1 view(left) | 0.715 | 0.181 | 3.412 | 9.834 |
> |4 views | **0.730** | **0.162** | **3.358** | **9.577** |
> |5 views | 0.695 | 0.202 | 3.613 | 9.502 |
>
> It is reasonable to assume that four views contain more 2D information and outperform one view.  Furthermore, the front view and left view do not have much distinction in terms of performance. However, the performance actually decreased with the additional top view. This could be due to the fact that the 2D information becomes more difficult to classify without the condition of the camera view. We did not pass along the camera information because it is difficult to estimate the camera view of in-the-wild videos, and injecting camera information could disrupt the structure of the text-to-2D model, making it difficult for the text-to-3D model to follow. In conclusion, the number of views serves as a practical hyperparameter that can be adjusted through enumeration experiments.
>
> We have provided a complete analysis in Appendix A, which we encourage you to refer to. Additionally, we were expecting that adding a top view would improve the performance, but the results showed the opposite. This has prompted us to think that more views do not necessarily lead to better results. A large gap between 2D information may complicate the easy-to-learn text-to-2D relationship. Therefore, the number of views serves as a hyperparameter in our CrossDiff model.
>
> **Q3: Is the framework aware of the input/target view?**
>
> No, we did not pass along the camera information. During training, each epoch, the 3D motion is paired with a random 2D projection. Our design is to extend our model to learn in-the-wild 2D pose while camera information is difficult to obtain. For further illustration, please refer to Q2 of the response to reviewer W5FS.
>
> **Q4: If the $x_{2D}\rightarrow\hat{x}_{3D}$ motion generation is linked to 3D pose estimation via lifting?**
>
> Yes, we also find that there is a similarity between $x_{2D}\rightarrow\hat{x}_{3D}$ motion generation and 2D motion lifting. [A] proves that 3D pose estimation via lifting is more accurate than the end-to-end method. The insight behind it might be that 3D motion is more difficult to extract features than 2D motion. So directly estimating 2D motion and lifting it works like specifying a feature that is easy to learn. Our method also considers learning the relation between 2D motion and text is easier, and we think this is helpful for data learning.
>
> **Q5: A video in the supp would be helpful.**
>
> As you requested, we have presented a video in the supplementary material to compare our results with other methods.
>
> [A] Martinez et al., A simple yet effective baseline for 3D human pose estimation, ICCV'17

---

> ### Author Response · Authors · 2023-11-22
> **Authors Response to Reviewer kGCp[2/2]**
>
> **Q6: What is the concrete conclusion we could draw from Figure 6(a)(the number of old PDF)?**
>
> The figure clearly indicates that enhancing R precision is achieved by increasing 3D sampling steps, whereas optimal FID performance necessitates a balanced combination of 2D and 3D sampling. This finding emphasizes the critical need for custom-tailored sampling strategies to meet specific performance criteria effectively. For a more intuitive understanding, we suggest refering the supplementary material, which includes a video providing visual comparisons.
>
> **Q7: Reformatting of Figure 6(the number in old PDF) and root-decoupled diffusion.**
> Thank you for pointing out the problem. We have moved both parts to the Appendix.
>
> Once again, we appreciate your enlightening thoughts and are open to any further questions or suggestions you may have.

---

### Official Review · Reviewer_eXxC · 2023-10-29

**Soundness:** 2 fair
**Presentation:** 3 good
**Contribution:** 2 fair
**Rating:** 5
**Confidence:** 4

**Summary:**

The authors propose a method to generative humanoid motion sequences based on textual description. The proposed method is described to take both 2D and 3D information as the generation prior, which is the focus of this paper compared to existing works. And the method can train with 2D motion data without 3D motion ground truth, making the application is more flexible under data constraints. And the key is a mixed but unified representation space sourced from either 2D or 3D data modalities.

**Strengths:**

- The idea of aligning the representation space for 2D and 3D space makes the application of the proposed method more flexible, especially when the 3D ground truth is limited.
- The experiments show that the performance of the proposed method is on par the state-of-the-art diffusion-based methods that use only 3d data for training.

**Weaknesses:**

- The authors claim an essential advantage of the proposed method as “to utilize 2D motion data without necessitating 3D motion ground truth during training, enabling the generation of 3D motion.”. However, through the experiments discussed in Sec 4.3, before training with 2D-only data, the model has been pretrained on the complete 3D motion dataset only. Therefore, the claim seems misleading to me. It makes good sense that when 3D data is available, by projecting the 3D to 2D representation, we can learn a joint representation space for both 2D and 3D space. WIth a language encoder, the motion space and the language space are connected, thus further making text-to-motion generation. By fine-tuning on new 2D-only data, the model learns new samples aligned under the 2D representation, thus extending the text-to-motion generation diversities. However, this practice can hardly be claimed as “using 2D motion data without 3D GT during training” in my opinion.
- The results showcased in Table 1 are not impressive.
- Ablation studies in Sec 4.4 lack a focus. If the claim to be proven is that additional 2D data can help boost the performance, the results support it but this is no surprising. Can authors elaborate more about the results and intentions in Sec 4.4?

**Questions:**

Please see my concerns listed above.

---

> ### Author Response · Authors · 2023-11-17
> **Authors Response to Reviewer eXxC**
>
> We would like to thank you for your valuable comments, and we have provided detailed answers to your concerns below:
>
> **Q1: The claim of “using 2D motion data without 3D GT during training”.**
>
> We acknowledge that the original description may have been misleading, and we want to emphasize that we have no intention of exaggerating the quality or significance of our work. We have revised the statement to "with a pretrained model, our approach accommodates using in-the-wild 2D motion data without 3D motion ground truth during training to generate 3D motion." We hope this clarification makes our attribution clear and removes any doubts.
>
> **Q2: The results showcased in Table 1 are not impressive.**
>
> We agree that our results are not impressive. However, we claim that our metrics are on the same order of magnitude as other state-of-the-art works. Additionally, we introduce a new metric measuring the FID of upper and lower body movements. With fewer joints measured, the model needs to perform more details to achieve higher performance. Our method outperforms other methods by a large margin with respect to FID-L, which proves our statement. For generation tasks, visualization sometimes matters more than metrics, and we suggest referring to Figure 4 and the video in the supplementary material for the effect of our method.
>
> **Q3: Ablation studies in Sec 4.4 lack a focus.**
>
> We appreciate your suggestion, and we have revised Section 4.4 accordingly. Our primary claim is that when training with paired 3D motion and text, projecting the 3D motion to 2D motion and building a connection between 2D motion and text can help boost performance. We have added an analysis of 3D and 2D motion in Appendix A and demonstrated that learning the connection between 2D motion and text is valuable. The control groups of "100% 3D" and "100% 3D + 100% 2D" in Table 2 prove our claim on a quantitative level. As our model is based on MDM, the visualization of "100% 3D" is nearly the same as MDM. From the comparison of our method and MDM in Figure 4 and the video, we can see the effect of learning 2D motion. Additionally, we have conducted more ablation studies in Section 4.4 and Appendix A to explore our model's capacity. Although our method may not be extremely surprising, we provide a simple and novel perspective on motion learning that is also enlightening in other research areas.
>
> Our secondary claim is that additional 2D data can help boost performance, as you mentioned. The additional 2D data indicates other 2D motion without ground truth 3D motion. The control groups of "50% 3D" and "50% 3D + 100% 2D" in Table 2 prove this claim on a quantitative level. Moreover, in Section 4.3, we demonstrate that learning 2D motion from other datasets helps with out-of-domain 3D motion learning.
>
> Once again, we appreciate your valuable comments and are open to any further questions or suggestions you may have.

---

> > ### Comment · Reviewer_eXxC · 2023-11-22
> > **Feedback**
> >
> > Dear authors,
> >
> > Thanks so much for your reply.
> >
> > Your revision and reply are very helpful in clarifying the intentions and contributions made in this paper. I read the revised version of the draft carefully and I think the current version is more self-contained. I recognize the contribution that lifting the knowledge about 3D motion from 2D-only data can help to improve 3D motion generation. But this is hard to be surprising and the quantitative evaluations don't show significant outperforming to me. Thus I am hesitating to adjust my rating. I will decide that afterward after a discussion with other reviewers.
> >
> > Also, the idea of using a shared feature encoder/extractor/representation for data from different domains has been studied for a very long time in the area of domain adaptation and transfer learning. You may want to read some papers following the line of "domain confusion"* in this area if you hasn't.
> >
> > *: "Deep Domain Confusion: Maximizing for Domain Invariance"

---

> > > ### Author Response · Authors · 2023-11-22
> > > **Authors Response to Reviewer eXxC**
> > >
> > > We greatly appreciate your valuable comments and recommendations, and we are grateful for the opportunity to further discuss our work with you.
> > >
> > > Firstly, we would like to emphasize the novelty and significant contributions of our study. Ours is the first research to investigate the role of 2D representation in 3D motion generation, and it also presents an innovative approach for denoising two modalities simultaneously. Our experiments have uncovered a previously unreported synergistic effect between 3D and 2D motion. We hope that our findings will help advance the field and promote a deeper understanding of motion generation. Moreover, we believe that the concept of utilizing 2D representation in diffusion models to complement information that might be overlooked in 3D space could benefit other research fields as well.
> > >
> > > In terms of quantitative evaluations, we would like to emphasize that high scores do not necessarily translate to natural-looking results in generative methods. Current evaluation metrics are based on the motion extractor proposed by Guo et al. [1], which may not capture all motion details and could potentially fail to assess the advantages of our approach accurately. As a result, we suggest measuring the Frechet Inception Distance (FID) for the upper and lower body separately, offering a more comprehensive evaluation.
> > >
> > > |Methods   |   FID-U  |   FID-L  |
> > > | -------- | -------- | -------- |
> > > |MDM       |   0.825  |   0.840  |
> > > |T2M-GPT   |   0.145  |   0.607  |
> > > |MLD       |   0.541  |   0.553  |
> > > |Ours      |   0.118  |   0.281  |
> > >
> > > Our method achieved the lowest FID for both upper and lower body parts, while previous works may have only excelled in one part and underperformed in the other. This demonstrates the uniqueness of our approach in comparison to previous methods.
> > >
> > > Finally, We appreciate your recommendation and have reviewed the suggested work. We concur that our approach involves learning a shared feature space, akin to domain adaptation. However, our method is designed to address a distinct problem. While domain adaptation primarily focuses on reducing the domain gap, our approach capitalizes on the capabilities of 2D representation to compensate for the limitations of 3D representation.
> > >
> > > [1] Guo et al. Generating Diverse and Natural 3D Human Motions From Text

---

### Official Review · Reviewer_W5FS · 2023-10-31

**Soundness:** 3 good
**Presentation:** 3 good
**Contribution:** 3 good
**Rating:** 6
**Confidence:** 3

**Summary:**

The paper introduces a Diffusion-based text-to-motion method. During training the method is trained on 3D as well as projected 2D pose representations. The method produces acceptable results when compared to SOTA.

**Strengths:**

While I find the major claim of the paper, that 2D human motion somehow contains more intricate motion details than 3D motion, hard to believe I find the results convincing. In particular, I find the application of learning novel motion modes from just 2D poses very interesting and relevant.

**Weaknesses:**

The major claim of the paper, that 2D human motion somehow contains more intricate motion than 3D human motion, is unconvincing as the 2D motion is strictly less “informative” as the 3D motion. It seems that the method requires a complex training strategy that aids the 3D Motion Encoder-Decoder to produce better results than SOTA. Can the authors comment on the capacity of their model in comparison to other SOTA methods? Could it be that the 2D skeletons provide regularization and help prevent overfitting a very large model?

Some architecture choices are not explained:
* In Mixture Sampling there seems to be no process to pass along the camera information. For example, what happens in Figure 3 if the 2D inputs would have been taken from another random camera, i.e. from the side? Would the 3D pose be rotated ?
* What is the purpose of the learnable token embeddings?
* Why is the text embedding added separately to the 3D and 2D encoder and not “jointly” in the “Shared Weights Encoder”?

Minor:
* Figure 6 is too small

**Questions:**

The authors should make clear what the % are in Table 2

---

> ### Author Response · Authors · 2023-11-17
> **Authors Response to Reviewer W5FS**
>
> We thank for your enlighting comments. Below are our response to your concerns.
>
> **Q1: The analysis of 2D and 3D motion.**
>
> As we have mentioned in **Response to All Reviewers**, when using only 3D representation, the generation model may struggle to relate text semantics to some body part movements with very small movement variations compared to others, which can lead to overlooking important motion details. This is because the model might focus on more dominant or larger movements within the 3D space, leaving subtle nuances underrepresented. Due to different view projections in 2D data, small body part movements can be magnified in certain projections, making them more noticeable and easier to capture. This helps the text-to-motion generation models to better associate text descriptions with a wider range of human body motion details, including subtle movements that might have been overlooked when relying solely on 3D representation.
>
> We have provided a complete analysis of 2D and 3D motion in Section 4.4.1, which should help in understanding why 2D motion contains more details. We propose the concept that 2D motion can help in 3D motion learning, regardless of the model used.
>
> **Q2: The effect of camera information in Mixture Sampling.**
>
> Firstly, all 3D motion is preprocessed to face the z+ direction at the starting frame. Therefore, the view of 2D motion would not rotate the 3D pose. This can be seen in Figure 6 (the number in the new PDF), where after learning with in-the-wild 2D pose (where we cannot guarantee the camera view), our 3D generation still faces z+ at the starting frame. Additionally, we plan to extend our work to learn abundant in-the-wild 2D human pose, where the camera information will be a weakness. Therefore, we do not pass along the camera information. However, we agree that learning with accurate camera information would improve the results.
>
>
> **Q3: What is the purpose of the learnable token embeddings?**
>
> We do not have any special purpose for learnable token embeddings. Our model is simple and nearly the same as the standard transformer model [1] when viewed in only one domain pipeline. As the original transformer decoder is inputted with the output embedding, we regard learnable token embeddings as a simple method to learn the output feature.
>
> **Q4: Why is the text embedding added separately to the 3D and 2D encoder and not “jointly” in the “Shared Weights Encoder”?**
>
> As mentioned in our response to Q1, we find that 2D motion contains different attention to the different joints. Therefore, we want to establish a direct connection between the 2D motion and text prompt and enforce the model to learn the connection between the different details and the text. If the text embedding is added to the "Shared Weights Encoder," the whole model is forced to learn the connection between 3D and 2D motion first and is only able to extract the feature of motion, which may result in the same output as extracting the feature from 3D motion alone. This would cause the remarkable details in the 2D domain to be fused and lose their value.
>
> In response to your feedback, we have moved the related experiments from Figure 6 (the number in the old PDF) to the Appendix to make room for other important experiments in the main body. Once again, we appreciate your valuable comments and are open to any further questions or suggestions you may have.
>
> [1]Vaswani et al. Attention Is All You Need

---

### Official Review · Reviewer_HDJQ · 2023-11-01

**Soundness:** 3 good
**Presentation:** 3 good
**Contribution:** 2 fair
**Rating:** 5
**Confidence:** 5

**Summary:**

This paper introduces a new framework for text-driven motion generation, with a primary focus on the simultaneous 2D and 3D motion denoising process. Both have separate input and output modules but share an intermediate transformer structure. Additionally, the authors have designed a new sampling method to better incorporate the knowledge from 2D into the 3D generation.

**Strengths:**

1. This paper attempts to simultaneously diffuse different forms of motion data, which is a fascinating direction and contributes to the research community. The ablation study also demonstrates its effectiveness.

2. The paper is well-written, making its content easily understandable for readers.

**Weaknesses:**

My primary concerns regarding this paper are related to the limited extent of experimental comparisons and analyses.

1. Some significant references are missed in this paper, such as ReMoDiffuse\[1\] abd Fg-T2M\[2\].

2. Some archiecture designs are not sufficiently evaluated. For example, why the authors choose to share the intermediate transformer. Quantiative results are required here.

3. The authors should provide user studies to quantatively evaluate the visual quality.


\[1\] Zhang et al. ReMoDiffuse: Retrieval-Augmented Motion Diffusion Model

\[2] Wang et al. Fg-T2M: Fine-Grained Text-Driven Human Motion Generation via Diffusion Model

**Questions:**

Please kindly refer to the weaknesses mentioned above.

---

> ### Author Response · Authors · 2023-11-17
> **Authors Response to Reviewer HDJQ**
>
> We would like to express our gratitude for your insightful comments and constructive feedback. We have carefully considered each concern and provide our responses below:
>
> **Q1: Add significant references.**
>
> We acknowledge our oversight and have taken steps to address it. Specifically, we have added the relevant work to both Section 2.1 and Section 4.
>
> **Q2: Effect of the shared-weights encoder.**
>
> In response to the reviewer's concern regarding the efficacy of the shared-weight encoder, we have conducted a evaluation, the results of which are presented as below.
>
> | Methods | R Precision(top 3) $\uparrow$ | FID $\downarrow$ | MM Dist $\downarrow$ | DIV $\rightarrow$ |
> | ----- | ------ | ----- | ----- | ---- |
> | w/o shared-weights encoder | 0.714 | 0.187 | 3.496 | 9.488 |
> | w/ shared-weights encoder | **0.730** | **0.162** | **3.358** | **9.577** |
>
> These empirical findings demonstrate that incorporating a shared-weight encoder significantly boost the the performance across several key metrics, including R Precision, Precision, FID, and DIV. This enhancement is not merely incremental but substantial, as evidenced by the quantitative results presented.
> Inspired by [1], we found that when learning with data from two modalities, extracting separate and fused feature layer-by-layer is more efficient. The shared-weights encoder serves as a fusing module, while the 3D/2D motion decoder acts as a separate decoder module. The goal is to ensure that the decoder layers follow the same extraction patterns as the shared-weight layers, rather than simply gaining deeper embeddings.
>
> For an in-depth analysis and discussion of these outcomes, we kindly direct the reviewer's attention to Section 4.4.3 of our revised manuscript, where these aspects are elaborated upon in detail. We hope this additional data addresses the concerns raised and further substantiates the efficacy of our proposed method.
>
> **Q3: Add the User study.**
>
> Thank you for the suggestion. We conducted a user study on motion performance, in which participants were asked two questions
> to assess the vitality and diversity of the motions. The first is "Which motion is more realistic and contains more details?" and the second is "Which generations are more diverse?". The results are presented as below.
>
> | Question | Ours vs MDM | Ours vs MLD | Ours vs T2M-GPT |
> | ----- | ----- | ----- | ----- |
> | q1 | 81% | 66% | 56% |
> | q2 | 60% | 58% | 69% |
>
> Our method produces more realistic and vivid generations while maintaining the diversity of the diffusion model. For visualization results, please refer to the supplementary material.
>
> Thank you once again for your valuable advice on the experimental settings. We greatly appreciate your feedback and are open to any further questions or suggestions you may have.
>
> [1] Xu et al. Building bridges between encoders in vision-language representation learning.

---

### Author Response · Authors · 2023-11-22
**Response to All Reviewers**

We gratefully acknowledge the concerns raised by most reviewers regarding how 2D representation could aid 3D human motion generation. Recognizing the critical importance of this aspect in our research, we have carefully revised our manuscript to address these concerns comprehensively:

1. **Refined Introduction (Section 1):**

   (a) We have provided a more detailed explanation of our statement regarding the limitations of using only 3D motion information during training. We have emphasized that relying solely on 3D representation may lead to overlooking important motion details, as the model might focus on more dominant or larger movements within the 3D space, leaving subtle nuances underrepresented.
   (b) We have expanded on our statement about the advantages of using 2D data representation in capturing the intricacies of human body movements from various viewing angle projections. We have explained that due to different view projections in 2D data, small body part movements can be magnified in certain projections, making them more noticeable and easier to capture. This helps the text-to-motion generation models to better associate text descriptions with a wider range of human body motion details, including subtle movements that might have been overlooked when relying solely on 3D representation.

2. **In-Depth Analysis and Discussion (Section 4.4.1):**

   We have provided an analysis of the movement variation comparisons of body parts between 3D and 2D representation. The results showcase that hand movement variation is small in 3D representation, while it can be magnified in 2D representation from a certain view projection. (Due to space limitation and reviewers' suggestions for additional experiments, we have moved the sections on "Comparisons of Sampling Method" and "Effect of Root-decoupled Diffusion" to the Appendix.)

3. **Visual Explanation Addition:**

   To further clarify the provided explanation, we have incorporated a video demonstration, which more clearly illustrates the benefits of incorporating 2D information into 3D human motion generation. https://1drv.ms/v/s!AtNfVe3pCI-nnXIzCcfXnQs9Jk6j?e=M8mE8c

We hope that these clarifications address your concerns and provide a better standing of our method and its contribution.

---

### Meta-Review · Area_Chair_1Wwn · 2023-12-10

**Metareview:**

This submission presents a novel framework for text-driven motion generation, notably addressing both 2D and 3D motion denoising processes. The paper has been reviewed by four experts in the field, receiving mixed reviews. All the reviewers commend the clarity and well-structured presentation of the paper. However, after reading the rebuttal, there are critical aspects highlighted by the reviewers that need to be addressed.

1) Firstly, there is a notable omission of discussion and comparison with several closely related works, such as 'ReMoDiffuse' and 'Fg-T2M'. Incorporating a thorough comparison and discussion of these works is essential to demonstrate the unique technical contributions.

2) Secondly, the reviewers express concerns regarding the depth of evaluation of the key architectural designs. A more comprehensive evaluation is necessary to establish the robustness and effectiveness of the proposed framework.

While the paper is undoubtedly well-written and introduces promising ideas, the consensus, in light of the reviewers' feedback, is to not recommend acceptance in its current state.  We strongly encourage the authors to consider and address these concerns.

**Justification For Why Not Higher Score:**

1) Several closely related works are not discussed and compared, e.g., ReMoDiffuse and Fg-T2M. 2) The key architectural designs are not sufficiently evaluated.

**Justification For Why Not Lower Score:**

N/A

---

### Decision · Program_Chairs · 2024-01-16

Reject